

# Promoting sustainable agriculture by exploiting plant growth-promoting rhizobacteria (PGPR) to improve maize and cowpea crops

Nadège Adoukè Agbodjato[1,2] and Olubukola Oluranti Babalola[1]

[1] Food Security and Safety Focus Area, Faculty of Natural and Agricultural Sciences, North West University, Mafikeng, North West, South Africa
[2] Laboratoire de Biologie et de Typage Moléculaire en Microbiologie (LBTMM), Département de Biochimie et de Biologie Cellulaire, Université d'Abomey-Calavi, Calavi, Benin

Corresponding author
Olubukola Oluranti Babalola,
olubukola.babalola@nwu.ac.za

## ABSTRACT

Maize and cowpea are among the staple foods most consumed by most of the African population, and are of significant importance in food security, crop diversification, biodiversity preservation, and livelihoods. In order to satisfy the growing demand for agricultural products, fertilizers and pesticides have been extensively used to increase yields and protect plants against pathogens. However, the excessive use of these chemicals has harmful consequences on the environment and also on public health. These include soil acidification, loss of biodiversity, groundwater pollution, reduced soil fertility, contamination of crops by heavy metals, *etc.* Therefore, essential to find alternatives to promote sustainable agriculture and ensure the food and well-being of the people. Among these alternatives, agricultural techniques that offer sustainable, environmentally friendly solutions that reduce or eliminate the excessive use of agricultural inputs are increasingly attracting the attention of researchers. One such alternative is the use of beneficial soil microorganisms such as plant growth-promoting rhizobacteria (PGPR). PGPR provides a variety of ecological services and can play an essential role as crop yield enhancers and biological control agents. They can promote root development in plants, increasing their capacity to absorb water and nutrients from the soil, increase stress tolerance, reduce disease and promote root development. Previous research has highlighted the benefits of using PGPRs to increase agricultural productivity. A thorough understanding of the mechanisms of action of PGPRs and their exploitation as biofertilizers would present a promising prospect for increasing agricultural production, particularly in maize and cowpea, and for ensuring sustainable and prosperous agriculture, while contributing to food security and reducing the impact of chemical fertilizers and pesticides on the environment. Looking ahead, PGPR research should continue to deepen our understanding of these microorganisms and their impact on crops, with a view to constantly improving sustainable agricultural practices. On the other hand, farmers and agricultural industry players need to be made aware of the benefits of PGPRs and encouraged to adopt them to promote sustainable agricultural practices.

## INTRODUCTION

Agriculture allows communities to guarantee their need for sustenance through the production of several types of crops. Maize (*Zea mays* L.) is one of the world's most widely grown, consumed, exported, and frequently rotated cereals (*Poole, Donovan & Erenstein, 2021*; *Rouf Shah, Prasad & Kumar, 2016*). Maize is an important staple food worldwide with nutritional, environmental, economic, and cultural (*Tanumihardjo et al., 2020*). Cowpea (*Vigna unguiculata*) is an important African food legume, well adapted to arid regions (*Ndungu et al., 2018*). It is an essential staple food for many people, especially in Asia and Africa. Its high protein and carbohydrate content characterizes it. Harvesting both cowpea leaves used as a vegetable and grains from the same plant (dual use) is an essential objective for farmers, enabling them to exploit the nutritional benefits offered by cowpea. The leaves of cowpea plants are consumed in a variety of traditional dishes or dried for use during the dry season (*Omomowo & Babalola, 2021*).

Growing food demand due to population growth has forced farmers to use chemical fertilizers and pesticides to increase crop yields and combat pathogens (*Babalola et al., 2021*). The massive use of these chemicals has undeniably increased yields and improved crop control. However, it has in turn polluted agro-ecosystems (water and soil) (*Riah et al., 2014*) with significant negative consequences on biodiversity and human health (*Igiehon & Babalola, 2018*; *Nicolopoulou-Stamati et al., 2016*). Since the 2000s, the adoption of so-called "industrial" agriculture, characterized by the intensive use of inputs, has generated significant environmental repercussions that it is now recognized as one of the main causes of overshooting "planetary limits", especially in terms of biodiversity (*Campbell et al., 2017*), hence continue in the same direction is no longer an option.

In addition, soil salinity is one of the world's major environmental challenges, transforming fertile land into unproductive land at a rate estimated at around 1–2% per year in arid regions. Soil salinization has rendered around 7% of the planet's land infertile, representing around 20–30% of the total arable land surface in general from natural geochemical processes and human-induced secondary salinization (*Hopmans et al., 2021*; *Rasool et al., 2013*).

Faced with this observation and to identify the axes to be developed to allow these ecosystems to render the expected services and satisfy the needs of local populations, numerous research works have been undertaken to optimize plant growth in stressed environments and to control specific soil components that could contribute to the reclamation of these degraded ecosystems (*Duponnois et al., 2013*). In this context, the use of beneficial microorganisms is gaining attention from researchers around the world to promote sustainable agriculture (*Agbodjato et al., 2022*; *Yadav & Sarkar, 2019*). Among these beneficial microscopic organisms are PGPRs that colonize plant roots. Thanks to their mechanisms of action, PGPR can improve crop yields under both stressful and non-stressful conditions (*Kálmán et al., 2023*; *Nadeem et al., 2010*; *Noumavo et al., 2016*).

PGPRs have demonstrated their ability to provide plants with essential nutrients to enhance their productivity, protect plants from stresses of abiotic and biotic origin, and prevent attacks by plant pathogens (*Agbodjato et al., 2021*; *Fasusi, Cruz & Babalola, 2021*). Several different genera of micro-organisms are currently used in agriculture worldwide (*Jalal et al., 2023*; *Prasad et al., 2019*).

PGPRs are increasingly being applied to sustainably improve the yield of keycrops such as maize and cowpea. The large-scale application of PGPR in cultivation could represent a success in reducing the need to use harmful pesticides and fertilizers, thus helping to preserve the environment (*Basu et al., 2021*). This review will provide a deeper understanding of the role of PGPRs, its potential to promote plant growth, and its application as a biological tool against pathogens and to mitigate the effects of environmental stressors.

## SURVEY METHODOLOGY

In order to ensure a comprehensive review of the literature and to achieve the objectives of the analysis, a thorough search of published articles on research terms related to the mechanisms of action of PGPRs, their effect on maize and cowpea growth and yield; the impact of chemical fertilizers on the environment, the biofertilizer market, the mode of action of beneficial rhizobacteria and their effect on maize and cowpea growth and yield was undertaken, following the methodology used by the authors below (*Agbodjato et al., 2021*; *Breedt, Labuschagne & Coutinho, 2017*; *Juby et al., 2021*; *Noumavo et al., 2016*; *Rocha et al., 2019*). Platforms such as Google Scholar, PubMed, ResearchGate, NCBI, *etc*. enabled us to access scientific publications more easily. All this information collected contributed to sustainable agriculture and food security. The results of this research were collected and grouped using EndNote bibliographic management software, enabling relevant articles to be identified and integrated into the context. However, an assessment of titles, abstracts, and conclusions in the literature was carried out to determine their relevance.

## IMPACT OF PESTICIDES ON SOIL FERTILITY

Conventional agriculture relies heavily on the intensive use of fertilizers and pesticides to increase crop yields and meet global food demand. However, a study by *Baweja, Kumar & Kumar (2020)* highlighted the many undesirable aspects associated with inorganic fertilizers and pesticides, which should not overlook. These chemicals can persist in the soil and environment for long periods, impacting various biotic and abiotic factors and leading to adverse effects on soil, microflora, surrounding organisms, the environment, and human health (*Baweja, Kumar & Kumar, 2020*). Prolonged intensive and indiscriminate use of these agrochemicals has had a negative impact on soil biodiversity, agricultural sustainability, and food security, with long-term detrimental effects on nutritional security, as well as on human and animal health (*Meena et al., 2020*). The same observations have been reported by *Hakim et al. (2021)* explaining that the uncontrolled and widespread use of chemical fertilizers constitutes a severe threat to the stability of an ecosystem and the sustainability of agriculture because these chemicals accumulate in the soil, leach into the water and are emitted into the air where they persist for decades, which constitutes a

serious threat to our environment. Several researchers have warned in their studies that the excessive use of pesticides could have a pernicious effect on the microflora that could potentially degrade soil fertility (*Aktar, Sengupta & Chowdhury, 2009*; *Glover-Amengor & Tetteh, 2008*). The repeated use of these complex chemicals such as fertilizers, herbicides, insecticides, fungicides, *etc.*, inevitably kills off the microbial life that is invaluable to the health of the soil ecosystem (*Shang et al., 2019*). Pesticides are chemical substances consisting mainly of herbicides, insecticides, and fungicides whose purpose is to control or kill pests. Insecticides have a more significant impact on soil microbes than herbicides, but less than fungicides, these complex chemicals (*Aktar, Sengupta & Chowdhury, 2009*). Insecticide use has a detrimental effect on soil microorganisms, which play an essential role in the transformation of nitrogen in soils. The amplification of this effect may depend on the specific type of insecticide used. Prolonged use of insecticides at high doses inhibits the nitrification process and disrupts the activities of the microbes involved in this process (*Gundi, Narasimha & Reddy, 2005*). The presence of certain insecticides such as monocrotophos, dichlorvos, lindane, malathion, chlorpyrifos, and endosulfan at concentrations above field-recommended rates leads to a reduction in the biochemical processes of nitrification and denitrification in soils (*Madhaiyan et al., 2006*). It should also be noted that fungicides also have an impact on the biochemical quality of the soil. Indeed, the fungicides mancozeb, trimorph, and benomyl inhibit enzymatic activity of soil dehydrogenase, urease, and phosphatase (*Milenkovski et al., 2010*). Herbicides diminish several beneficial biochemical processes regulated by soil microorganisms, as well as the enzymatic reactions that play an essential role in maintaining or improving soil health, as highlighted by *Kinney, Mandernack & Mosier (2005)*. Herbicides also influence soil enzyme activities thus affecting the biological index of soil fertility and biological functions in the soil profile (*Antonious, 2003*). When used at concentrations higher than recommended in the field, the herbicides alachlor and atrazine have a negative impact on the function of bacteria essential for ammonification and dehydrogenase activity in the soil (*Demanou et al., 2006*). Therefore, the abuse of chemical fertilizers and pesticides leads to a decrease in the number of beneficial microorganisms present in the soil, which reduces soil fertility (*Fox et al., 2007*).

## RHIZOSPHERE MICROBIOME

### The rhizosphere, source of beneficial microorganisms for agriculture

According to German Lorenz Hiltner, the rhizosphere is considered to be the soil zone that surrounding the root of a plant, which is influenced by the soil, and has high microbial activities (*Hiltner, 1904*). Indeed, the rhizosphere is composed of three zones, such as: the endorhizosphere, the rhizoplane, and the ectorhizosphere (*McNear, 2013*). According to the observations of *Hassan, McInroy & Kloepper (2019)*, the rhizosphere is a specific ecological environment where beneficial bacteria compete with other microorganisms for access to organic carbon compounds. A number of these beneficial rhizobacteria, having colonized plant roots, are also able to multiply endophytically within root tissues.

Several microorganisms such as bacteria, archaea, fungi, *etc.* inhabit the rhizosphere interacting in various ways with their host plants (*Odelade & Babalola, 2019*). Figure 1

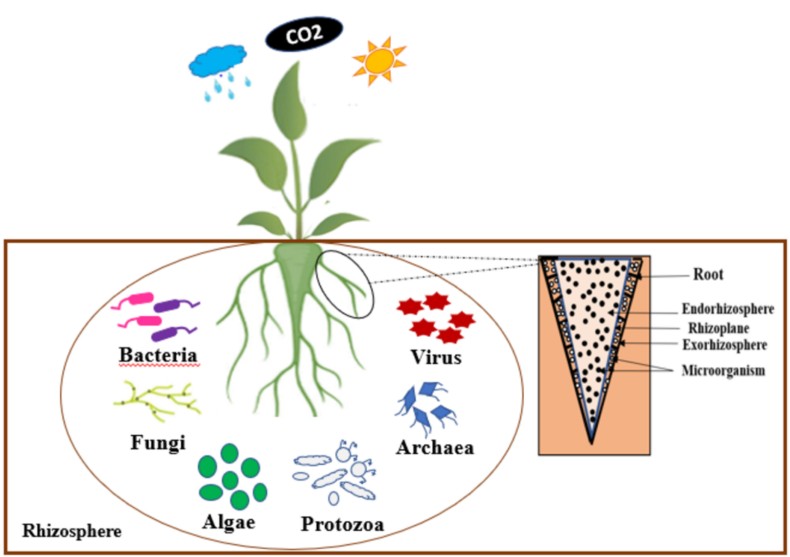

**Figure 1 Schematic representation of rhizosphere.**

presents an illustration of the rhizosphere. The interactions between the micro-organisms and their host plants present in this rhizosphere improve the growth and development of these plants. The survival of these microorganisms in the rhizosphere depends on their ability to invade the plant's root and out-compete native organisms (*Babalola et al., 2021*).

The composition of microbial communities in the rhizosphere is influenced by various factors, including host plant variety, plant growth stage, root zone, and environmental constraints (*Marschner, Crowley & Rengel, 2011*). Microbial groups may be composed of different species with the same modes of action but with different resistance to environmental stress conditions and plant cultivars (*Berg & Smalla, 2009*). PGPR is among the most important components of the soil microbiota. They enhance plant growth through their ability to adapt under stress conditions to interactions between plants and microorganisms that can negatively or positively affect the function of plant roots and to improve their growth induce several plant physiological processes, such as seed germination, photosynthetic rates, root growth and branching, yield (*Berg, 2009*; *Venkateshwaran et al., 2013*). These PGPR also influence the stability of soil aggregates, soil pH, pathogen activity, *etc.* (*Agbodjato et al., 2021*; *Berg, 2009*; *Venkateshwaran et al., 2013*).

The use of rhizosphere engineering in conjunction with the multiple activities of PGPRs offers wide application possibilities, for crop fertilization and the development of sustainable, environmentally-friendly agriculture (*Hakim et al., 2021*).

## Rhizodeposition and its interaction with PGPR in the rhizosphere

A complex interaction occurs between soil, plant, and microorganisms during the plant development cycle. This interaction is strengthened when the microbial community in the rhizosphere is stimulated by the release of rhizodeposits by plants during rhizodeposition (*Tabassum et al., 2017*). Rhizodeposition is defined as a wide range of substances released

by plant roots, including water-soluble exudates, lysates, fine decomposing roots, secretions of insoluble compounds, gases such as $CO_2$ and ethylene (*Cheng & Gershenson, 2007*).

Root exudation is the dynamic release of organic carbon compounds into the soil by plant roots, playing a crucial role in rhizodeposition (*Hütsch, Augustin & Merbach, 2002*; *Nguyen, 2009*). Various microorganisms depend on these rhizodeposits, establishing a gradient of interactions ranging from plant growth promotion to parasitism (*Tabassum et al., 2017*). However, rhizodeposition processes are influenced by many biotic (plant species, photosynthesis, root architecture, mycorrhization, and nodulation) and abiotic (temperature, moisture, rooting depth, soil texture) factors (*Jones, Hodge & Kuzyakov, 2004*).

Within the rhizosphere, is a reciprocal benefit relationship between roots and microorganisms. In this interaction, the two parties establish an exchange of nutrients essential to their respective survival. However, several types of relationships can develop within this rhizosphere depending on the abundance of nutrients in the soil, such as associative, symbiotic, neutral, or parasitic relationships (*Parmar & Dufresne, 2011*). According to the observations of *Kuzyakov & Xu (2013)*, rhizodeposits provide an energy source for soil microorganisms, promoting the solubilization of organic nitrogen and other nutrients present in soil organic matter. In addition, various types of PGPR may react in distinct ways to nutrients present in the soil rhizosphere, secreting multiple compounds. Some of these compounds secreted by PGPRs are known to suppress plant pathogens, while others promote plant growth (*Hassan, McInroy & Kloepper, 2019*). In addition to their capabilities above, PGPRs also compete with other microorganisms for nutrient acquisition in the rhizosphere. Thus, cells and exudates from the root zone are essential for plant growth and inhibition of a variety of soil pathogens (*Hassan, McInroy & Kloepper, 2019*). In conclusion, rhizodeposition plays a very important role in the rhizosphere ecosystem, providing an essential carbon source for PGPR and other beneficial microorganisms. This interaction is crucial for plant growth and health, improved soil structure, soil biodiversity, specific interactions between plants and PGPR and agricultural sustainability. By promoting these interactions, we can reduce dependence on chemical fertilizers and pesticides, as well as the environmental impact of agriculture. Research must continue in this field to better understand these interactions in order to develop more sustainable and efficient agricultural practices.

## ROLES OF SOIL ENZYMES IN THE RHIZOSPHERE AND THEIR CONTRIBUTION TO THE IMPROVEMENT OF PGPR

Enzymes present in the soil play a crucial role in energy transfer by breaking down soil organic matter and facilitating nutrient cycling, directly impacting soil fertility (*Nosheen et al., 2018*). As such, they act as essential catalysts for the decomposition of soil organic matter, and nutrient cycling and strongly influence on soil fertility, energy conversion, environmental quality, and agronomic productivity (*Rao et al., 2017*). Enzymes produced and released by roots and microorganisms play a crucial role in modifying nutrient availability in the rhizosphere (*Gianfreda, 2015*). Moreover, according to this author,

rhizosphere enzymes are generally classified into two main categories: cytosolic enzymes and extracellular enzymes.

Enzymes present in the cytosol, in association with cellular debris, likely play an essential role as a source of carbon, nitrogen, and reducing equivalents for the growth of microbial communities. Among the main soil enzymes involved in nutrient cycling are: (i) urease, which converts urea to ammonia and then to nitrite, thus facilitating the plant's easy access to nitrate ions (*Maithani et al., 2017*), (ii) phosphatase, which plays a crucial role in phosphate solubilization and the release of inorganic phosphate (*Satyaprakash et al., 2017*), and (iii) invertase, which plays an active role in the carbon cycle by catalyzing the hydrolysis of sucrose into glucose and fructose (*Wang et al., 2013*).

Plant roots or microorganisms intentionally express extracellular enzymes in the external environment. They perform either a protective function (oxidoreductases) by converting toxic soluble phenolic metabolites into insoluble polymeric products, or a degradative function (hydrolases and oxidoreductases) by hydrolyzing or oxidizing polymeric substrates such as lignin, humic acids or phenols for metabolic purposes (*Gianfreda, 2015*). Microorganisms releasing lytic enzymes (usually extracellular) have the initial role of degrading high molecular density polymers, which can also result in the direct suppression of phytopathogenic fungi (*Agbodjato et al., 2018*; *Egamberdieva et al., 2011*).

As a general rule, increased rhizosphere enzyme activity contributes to greater functional diversity in the microbial community.

# PLANT GROWTH-PROMOTING RHIZOBACTERIA (PGPR)

## Definition

The microorganisms that colonize plant roots in the rhizosphere and promote their growth are known as PGPR. This terminology was first introduced by *Kloepper (1978)*, who defined PGPRs as microorganisms capable of colonizing plant roots and enhancing their growth and yield. Root colonization by PGPRs is considered an essential step in promoting plant growth. Rhizobacteria spread from an inoculum source, such as seed treatment, to the actively growing root zone, where they multiply or develop in the rhizosphere. This process is known as root colonization (*Hassan, McInroy & Kloepper, 2019*).

## Mechanisms in promoting plant growth

PGPR is known to improve crop yield and plant protection. Studies on PGPR have attracted increasing scientific interest in recent decades. Through direct or indirect mechanisms, the PGPR stimulates the growth and development of plants (*Ahemad & Kibret, 2014*). Direct mechanisms refer to the traits that directly promote plant growth (Fig. 2). This mechanism facilitates the uptake of nutrients from the soil. Indirect mechanisms protect plants against phytopathogens (Fig. 2). Different PGPR possess one or both characteristics, and their activities may vary depending on the environmental and soil conditions (*Olanrewaju, Glick & Babalola, 2017*).

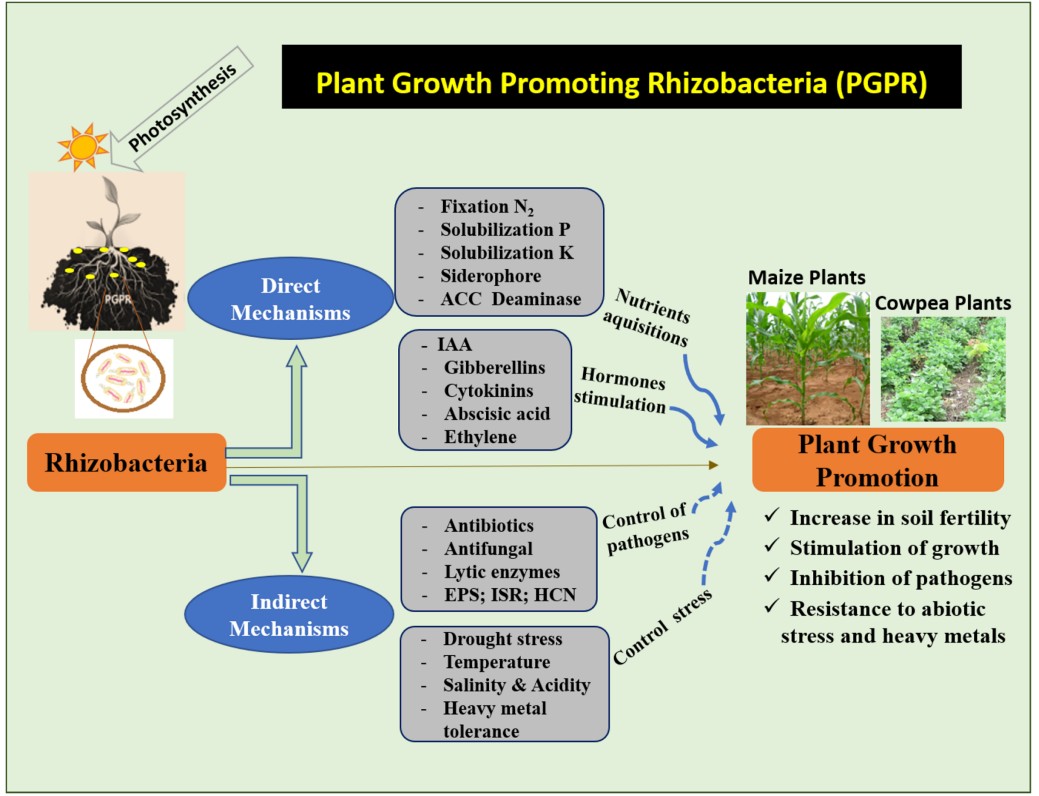

**Figure 2 Mechanism of action of PGPR to increase maize and cowpea plants.** (N$_2$) Nitrogen, (P) phosphate, (K) potassium, (ACC) 1-aminocyclopropane-1-carboxylate, (IAA) indole-3-acetic acid, (EPS) exopolysaccharides, (ISR) induced systemic resistance, (HCN) hydrogen cyanide.

### Direct mechanisms

#### Nitrogen fixation

Nitrogen (N$_2$) is an essential macronutrient for the plant. Atmospheric nitrogen (N$_2$) is present in large quantities in the air and cannot be directly assimilated by plant roots. However, due to the biological fixation of nitrogen by microorganisms, this nitrogen (N$_2$) is transformed into ammonia (NH$_3$) following an enzymatic reaction catalyzed by the enzyme nitrogenase which the roots can easily absorb. Microorganisms capable of biologically fixing atmospheric nitrogen (N$_2$), commonly known as "diazotrophs", are able to do so in association with plant roots. Bacteria can achieve biological nitrogen fixation in a symbiotic or non-symbiotic relationship. In the former case, this occurs with legumes or other higher plants such as alder (*Alnus* spp). In the second case, it is carried out by heterotrophic or autotrophic bacteria living in the plant rhizosphere (*Aasfar et al., 2021*; *Sickerman, Hu & Ribbe, 2019*). The use of biological PGPR inoculants, which can fix nitrogen, on crops has benefits such as improved growth, and disease management. Nitrogen-fixing PGPR has been reported as a valuable source of nitrogen for sustainable crop production and maintenance of soil fertility (*Singh et al., 2017*). Examples of diazotrophic bacteria living freely in the rhizosphere and providing available nitrogen to many plants, thus promoting their growth, include the genera *Azotobacter, Bacillus,*

*Anabaena, Azospirillum, Klebsiella, Paenibacillus* and *Rhodobacter* (*Grobelak, Napora & Kacprzak, 2015*). Numerous symbiotic bacteria have reported, among which are identified the genera Frankia spp, associated with some dicotyledonous species, *Rhizobium* spp, associated with legumes and other *Azospirillum* spp associated with grasses (*Pankievicz et al., 2019*; *Singh et al., 2017*).

**Phosphate (P) solubilization**

Phosphorus (P) plays an essential role in the growth and development of plants. It is available in large quantities in soils in organic and inorganic forms (*Ahemad & Kibret, 2014*). Generally, not more than 5% of these two available forms are soluble so the plants cannot be directly absorb them. In this case, the PGPR strains can solubilize these phosphates in an assimilable form for good absorption by the roots of the plant. The solubilization and mineralization of phosphorus depend on the action of the soil bacteria (*Liang et al., 2019*). The solubilization of inorganic phosphates occurs through the production of dissolving mineral compounds such as organic acids, siderophores, protons, hydroxyl ions, and $CO_2$ (*Sharma et al., 2013*). These organic acids, along with their carboxyl and hydroxyl ions, chelate cations or reduce pH, enabling the release of phosphorus (P) (*Usha & Padmavathi, 2012*). The mineralization of organic phosphorus, occurs through the hydrolysis of phosphoric esters by the action of phosphatases (*Alori, Glick & Babalola, 2017*). PGPR uses different methods to transform insoluble phosphates present in the soil into soluble forms, thus becoming assimilable by plants. PGPRs primarily employ the following methods: (i) the release of complex compounds and inorganic solvents such as hydroxyl ions, organic acid anions, $CO_2$, and protons; (ii) the release of extracellular enzymes; and (iii) the release of phosphate upon substrate degradation (*Oleńska et al., 2020*). Bacteria from the genera *Bacillus, Pseudomonas, Serratia*, and *Enterobacter* have been reported to solubilize insoluble phosphate compounds, promoting plant growth and yield (*Kumar et al., 2008*; *Stein, 2005*).

**Potassium (K) solubilization**

Potassium (K) is an essential macronutrient for plant growth. However, concentrations of soluble potassium in soil are generally deficient, and over 90% of potassium in soil is insoluble, making potassium unavailable to plants (*Parmar & Sindhu, 2013*). However, many microorganisms, including certain bacterial species that establish mutual relationships with plants, are able to solubilize potassium in the soil (*Setiawati & Mutmainnah, 2016*). Rhizobacteria mainly use the production of organic acids, inorganic acids, and protons as their main mechanisms for solubilizing potassium (*Maurya, Meena & Meena, 2014*; *Meena et al., 2015*). Studies have reported that PGPRs capable of solubilizing potassium, such as *Paenibacillus* sp., *Bacillus edaphicus*, *Acidothiobacillus* and *Pseudomonas*, release potassium in a manner accessible to plants from insoluble sources present in soils, thus promoting optimal plant growth (*Liu, Lian & Dong, 2012*).

**Siderophores production**

Siderophores are low molecular weight compounds that play a crucial role in microbial-plant interactions in the rhizosphere (*Shanmugaiah et al., 2015*). In general, the iron (Fe) available in the soil is not directly assimilable by plants because it is trivalent hydroxide Fe3+. molecules of siderophores are therefore produced and secreted by

bacteria, which facilitates the assimilation of iron by plants. Catecholates/phenolates, hydroxamates, and carboxylates are the different groups of siderophores that exist on the chemical level. They are classified according to their chemical role in iron chelation. The siderophores produced by PGPR have a strong affinity for iron, enabling them to sequester even low concentrations of iron in the soil (*Saha et al., 2016*). In their study, *Kumar et al. (2018)* highlighted that the inoculation of siderophore-producing PGPRs, and the use of their consortium positively impacted wheat growth and development. The production of siderophores by PGPRs plays an essential role in biological control, as it restricts the availability of iron to harmful plant pathogens, thus keeping them away from the sources of iron needed for their development (*Shanmugaiah et al., 2015*). The use of iron-chelating hydroxamate siderophores produced by the *P. aeruginosa* strain was associated with growth inhibition of the pathogens *R. solani* and *C. gloeosporioides* in chili plants, as demonstrated in the study by *Sasirekha & Srividya (2016)*. In addition, the presence of PGPR in the soil promotes the production of siderophores, enabling plants to reduce heavy metal toxicity in stressful environments (*Ayangbenro & Babalola, 2017*).

**ACC deaminase**

Another important direct mechanism is the exhibition of the enzyme 1-aminocyclopropane-1-carboxylate (ACC) deaminase activity by PGPR. PGPR-producing ACC enzyme is capable of hydrolyzing ACC to α-ketobutyrate and ammonia, which could reduce the high level of ethylene in plants. Indeed, the high level of ethylene in plants could decrease plant growth and lead to their death. This is the case for the ethylene content in plants (*del Carmen Orozco-Mosqueda, Glick & Santoyo, 2020*; *Singh et al., 2015*; *Tiwari, Duraivadivel & Sharma, 2018*). In stressful environmental conditions, ACC oxidase and ACC synthase activity in plants increase, leading to increased ethylene production. PGPRs that produce the ACC deaminase enzyme can help plants mitigate the negative effects of stress by promoting their adaptation and survival in these stressful conditions. This is achieved by reducing the amount of ACC available for conversion to ethylene, enabling plants to better cope with environmental stress (*del Carmen Orozco-Mosqueda, Glick & Santoyo, 2020*). A study by *Vocciante et al. (2022)* revealed an interaction between ACC deaminase activity and IAA action in the plant. Indeed, the action of the IAA on the plant could stimulate ACC synthase transcription, which would help lead to an elevation of ethylene levels in the plant. In the plant, the high level of ethylene in the plant induces an inhibitory response to the production of IAA, which leads to an inhibition of the growth of the plants. Several studies have demonstrated that the application of PGPRs producing the ACC deaminase enzyme confers on plants increased tolerance to various environmental stresses (*Pourbabaee et al., 2016*; *Saikia et al., 2018*). Interestingly, the ACC-deaminase gene has been signaled in some bacterial strains, which include *Burkholderia*, *Achromobacter*, *Azospirillum*, *Agrobacterium*, *Ralstonia*, *Rhizobium*, *Pseudomonas* and *Enterobacter* (*Gamalero & Glick, 2015*).

**Production of phytohormones (auxin, gibberellin, cytokinin, ethylene, abscisic acid)**

Auxin, ethylene, cytokinin, gibberellin, and abscisic acid are the five main groups of plant hormones recognized by botanists. The production of growth phytohormones produced by PGPR influence seed germination, and root development facilitating good nutrient uptake, resulting in good vascular tissue and shoot development, good flowering,

and full plant development (*Adeleke & Babalola, 2021*; *Antar et al., 2021*). These phytohormones also regulate hormone levels, influence photosynthetic processes favorable to plant growth and development, and activate defense responses against pathogens (*Backer et al., 2018*). These phytohormones were further discussed below:

Auxin, particularly indole-3-acetic acid (IAA), is a phytohormone present in plant-associated bacteria, playing a crucial role in various cellular and developmental processes throughout the bacterial life cycle. IAA is mainly produced in the rhizosphere of healthy plants and is involved in interactions between plants and bacteria. Its production is considered one of the key mechanisms promoting plant growth and development, notably by stimulating root formation and growth (*Grobelak, Napora & Kacprzak, 2015*; *Luo, Zhou & Zhang, 2018*; *Poveda et al., 2021*). In their study, *Bashan, Holguin & De-Bashan (2004)* showed that IAA production by *Azospirillum*, *Agrobacterium*, *Pseudomonas*, and *Erwinia* increases root hairs, root length, branching, and root area.

Cytokinins (CKs) play an essential role in the development of the vascular system, embryogenesis, nodule formation, and the response of plants to environmental variations. Their action is decisive in these processes (*Hönig et al., 2018*). CKs play a special role in interacting with auxin to counteract the aging process in plants. They act by modifying protein levels and promoting chlorophyll production in leaves, leading to a reduction in the yellowing of plant leaves (*Guo et al., 2021*).

Gibberellins (GA) produced by plant-associated microorganisms play a critical role in plant growth and development (*Nett et al., 2017*). Gibberellin controls the growth and development of the plant life cycle. Pre- and post-gibberellin control differs among cells and tissues, growth levels, environmental conditions, and plant species (*Rizza & Jones, 2019*).

Ethylene (E) is a growth phytohormone produced naturally by most plants and can induce multiple physiological changes at the molecular level in plants (*Shaharoona et al., 2011*). Ethylene production is essential role in apical meristem function and root development in plants. Depending on the particular plant species, ethylene concentration can affect plant growth, either stimulating or inhibiting it (*Vandenbussche & Van Der Straeten, 2012*). Certain plant-associated microorganisms can up regulate the enzyme ACC deaminase, which is responsible for the degradation of ACC, the precursor of ethylene in plants under stress. Under stress conditions, inoculation with these microorganisms can enhance plant growth and development, thereby reducing stress-induced ethylene production in the plant (*Shaharoona et al., 2011*).

Abscisic acid (ABA) is a plant hormone synthesized by plants in response to abiotic stress. Its role is to activate the expression of genes involved in stress resistance (*Sah, Reddy & Li, 2016*). ABA significantly impacts plant defense against bacterial and fungal pathogens and can induce different virus resistance mechanisms at any time of induction (*Alazem & Lin, 2017*). ABA can also affect seed germination inhibition, plant senescence induction, alongside fruit and leaf abscission (*Munemasa et al., 2015*).

### Indirect mechanisms

Indirect mechanisms occur when the microorganisms counteract or prevent the damaging effects of pathogens on plants by synthesizing antibiotics and cell wall degrading lytic enzymes (*Adeleke et al., 2022*).

#### Antibiotics production

The main action used by PGPRs is to synthesize one or more antibiotics to counteract the harmful effects of plant pathogens. This mechanism involves inhibiting the growth of phytopathogens through the production of secondary metabolites with antifungal and/or antibiotic properties (*Noumavo et al., 2016*), thus sustaining plant and soil health. In their review, *Prashar, Kapoor & Sachdeva (2013)* highlighted that PGPRs can produce antibiotics such as lipopeptides, polyketides, and antifungal metabolites, which have pathogen-suppressive properties. Several researchers have also demonstrated that certain PGPR strains have promising potential as biocontrol agents to combat plant pathogens (*Abdelmoteleb & González-Mendoza, 2020*; *Agbodjato et al., 2018*; *Viljoen et al., 2019*). However, *Glick (2015)* mentioned in these works that an antibiotic known to control one pathogen on the plant may also be without a positive effect on another pathogen on the same plant. In this case, the bacterium synthesizing the antibiotic may show variable action in the field under different conditions. *Riaz et al. (2021)* stated that *Bacillus thuringiensis* (Bt) produces pest-killing proteins, which is why it is widely used to protect plants against pests. The intervention of biological agents such as PGPRs coupled with proper plant nutrition can improve the agricultural importance of different plant species (*Jalal et al., 2023*). Biopesticides incorporating PGPRs offer a sustainable, environmentally-friendly approach to combating plant pests and diseases.

#### Induced systemic resistance (ISR)

Induced systemic resistance (ISR) is the defense capacity that plants acquire when stimulated adequately by various biological entities of rhizobacteria (*Conrath et al., 2015*; *Mariutto et al., 2011*). *Conrath et al. (2015)* explain PGPR-induced systemic resistance (ISR) as the defensive ability of plants in response to various pathogens. ISR manifests itself through different defense mechanisms, such as an increase in β-1,3 glucanase, chitinase, and peroxidase activity, as well as the accumulation of low-molecular-weight antimicrobial substances such as phytoalexins, and the formation of protective biopolymers such as lignin, callose and glycoproteins (*Archana et al., 2011*). Systemic resistance induced by exogenous chemical agents and pathogenic organisms is often referred to as systemic acquired resistance (SAR), while the protection conferred on plants by PGPRs is generally referred to as PGPR-induced systemic resistance (*Adeleke & Babalola, 2022*; *Kloepper, Tuzun & Kuć, 1992*). Using PGPR strains as an essential component of plant defense may increase their applicability and offer a practical approach to immunization (*Annapurna et al., 2013*; *de Andrade et al., 2023*).

#### Lytic enzymes production

Another important feature of PGPRs in the biocontrol of plant pathogens is the production of lytic enzymes (usually extracellular) that degrade pathogen cell walls (*Agbodjato et al., 2018*; *Egamberdieva et al., 2011*; *Kobayashi et al., 2002*). Rhizosphere

microbes produce hydrolytic enzymes that inhibit the growth of pathogens by degrading their cell walls, proteins, and DNA, making them safer, sustainable, and environmentally friendly biocontrol agents than chemical fungicides (*Jadhav & Sayyed, 2016*). Some PGPR produces enzymes such as chitinases, proteases, lipases, phosphatases, cellulases, β-glucanases, *etc.* that are capable of lysing the cell walls of many plants' pathogenic fungi (*Lanteigne et al., 2012*; *Mamta et al., 2012*). Research by *Karthika, Midhun & Jisha (2020)* has also shown that certain enzymes such as proteases, amylases, β-1,3-glucanases, cellulases, lipases, and xylanases produced by strains of *Bacillus cereus* and *Bacillus cepacia* are capable of degrading the cell walls of various pathogenic microorganisms present in the soil. These hydrolytic enzymes secreted by PGPR can induce direct elimination of the activities of plant pathogens. Bacteria such as *B. subtilis, B. cereus, B. thuringiensis* and *S. marcescens* can secrete hydrolytic enzymes to control plant pathogens (*Jadhav & Sayyed, 2016*).

**Competition**

Another mechanism unique to PGPR is their ability to compete as a biocontrol agent competes with and, in most cases, out-competes plant pathogens for either nutrients or binding sites in the rhizosphere of the host plant (*Etesami & Maheshwari, 2018*). This competition results in the blockage of phytopathogens from binding to plants until available nutrients are depleted, thus making it relatively impossible for them to increase and infect the plant. It has been reported that PGPR inoculants exhibit antifungal activity and can compete with pathogenic fungi such as *Fusarium oxysporum* (*Ambrosini et al., 2015*). In addition, *Bacillus megaterium's* competitiveness in promoting tomato plant growth has also been reported (*Lindström & Mousavi, 2020*).

**Hydrogen cyanide (HCN) production**

Hydrogen cyanide (HCN) is a volatile secondary metabolite that can inhibit the growth of pathogenic microorganisms and reduce negative effects on plant growth and development (*Siddiqui et al., 2006*). *Rijavec & Lapanje (2016)* also noted that HCN is recognized as a biocontrol agent due to its toxicity to plant pathogens. HCN can impact plant establishment or inhibit disease development, giving it strong potential to control of bacterial diseases in plants (*Lanteigne et al., 2012*). Several bacterial genera can produce HCN, notable among them include *Alcaligenes, Bacillus, Aeromonas, Rhizobium* and *Pseudomonas* (*Alemu, 2016*).

**Exopolysaccharides (EPS) production**

Exopolysaccharides (EPS) produced by PGPR enter the soil aggregates and modify their porosity (*Alami et al., 2000*). Bacterial EPS also combat salinity stress by limiting the amount of sodium in the soil (*Upadhyay, Singh & Singh, 2011*). Bacterial EPS, under water stress conditions, limits or delay the soil's desiccation. On the other hand, in case of rainfall and flooding in the soil, the EPS block the dispersion of clays in the soil (*Henao & Mazeau, 2009*). Plants inoculated with PGPR bacteria showed tolerance to water stress through improved soil aggregation (*Bashan, Holguin & De-Bashan, 2004*) and soil structure (*Sandhya et al., 2009a*). The involvement of the EPSs produced by PGPR under salt stress and drought conditions was crucial in improving *Helianthus annuus* yield (*Tewari & Arora, 2014*). The production of EPS by PGPRs is a crucial feature that helps to enhance

plant development during periods of drought. EPS enables the formation of hydrophilic biofilms around plant roots, protecting them from drying factors present in the soil (*Rolli et al., 2015*).

**Control abiotic stress and heavy metals by PGPR**

**Control abiotic stress**

Climate change has increased the effects of environmental stressors and thus affected agricultural production worldwide. Indeed, the high quantities of soil and low water resources in soils, lead to a decrease in agricultural productivity, also leading to the abandonment of the affected lands which indirectly affects the food security of the population (*Adeleke & Babalola, 2020*; *Ilangumaran & Smith, 2017*). In this context, plants are often influenced by environmental stresses, such as biotic stress (fungi, bacteria, and viruses) and abiotic stress (pH, temperature, drought, salinity, contaminants, *etc.*), causing adverse effects on plant survival (*Islam et al., 2016*). Some PGPRs can increase crop yield by improving tolerance to abiotic stressors and phytopathogens. *Kálmán et al. (2023)* also mentioned in their study that PGPRs help reduce the negative effects of drought and stressful thermal conditions on plants. The synthesis of the ACC deaminase enzyme is the most studied among the different PGP traits involved in the drought tolerance of plants. PGPR maintains ethylene levels below inhibitory levels through the production of ACC deaminase, which leads to normal root growth by avoiding excess auxin and slowing down senescence under drought conditions (*Marasco et al., 2013*). *Sandhya et al. (2009b)* mentioned that *Pseudomonas* sp. inoculation of maize increased solutes and modified antioxidant status in drought conditions. Studies by *Kour et al. (2020)* showed that inoculation with the drought-tolerant, phosphorus-solubilizing strain *Pseudomonas libanensis* EU-LWNA-33 improved wheat growth and phosphorus uptake under drought conditions. According to studies by *Jochum et al. (2019)*, inoculation of maize and wheat seedlings with *Bacillus* sp. 12D6 and *Enterobacter* sp. 16i delayed the onset of drought symptoms. The same authors also found that inoculation of wheat seed with *Bacillus* sp. (12D6) led to an increase in root length, while inoculation of maize seed with *Bacillus* sp. 12D6 and *Enterobacter* sp. 16i led to an increase in root length and surface area.

**Heavy metals**

Heavy metals, when discharged as effluents by industries, threaten the environment and public health. These heavy metals, usually consisting of mercury (Hg), cadmium (Cd), lead (Pb), arsenic (As), chromium (Cr), and antimony (Sb), affect the growth and yield of crops. Heavy metals are widespread pollutants, non-degradable, and therefore persistent in soils and a real environmental problem (*Omomowo & Babalola, 2021*; *Wuana & Okieimen, 2011*). Generally, heavy metals are toxic to plants which affect their growth. Nevertheless, some bacteria can neutralize the toxicity of these metals using negatively charged functional groups present in their cell wall. These functional groups enable interactions with positive metal ions, a phenomenon is known as metal biosorption (*Syed & Chinthala, 2015*). Bioremediation is one of the techniques used by microorganisms whereby their products are used to remove or immobilize contaminants in the environment without affecting the plant (*Azubuike, Chikere & Okpokwasili, 2016*). *Ojuederie & Babalola (2017)* also mentioned in their study that one of the important tools to restore eroded areas and

heavy metal-contaminated sites is bioremediation. Rhizoremediation is, the use of rhizobacteria to remediate or immobilize these environmental contaminants. PGPRs are considered beneficial agents capable of enhancing plant growth by eliminating or reducing pollutants in the environment (*Koul et al., 2019*). Abiotic stresses, such as heavy metals, lead to ethylene and stress hormones synthesis in plants. In high ethylene concentrations, root growth and proliferation are negatively affected in soils contaminated with heavy metals. However, microorganisms producing the ACC deaminase enzyme are beneficial in promoting plant growth under conditions of heavy metal contamination. Indeed, microorganisms produce the ACC deaminase enzyme, which regulates the concentration of ACC, the ethylene precursor, thereby helping plants survive under stressful conditions. This ability of microorganisms to metabolize ACC is beneficial to plants and has been demonstrated in studies such as those by *Mishra, Singh & Arora (2017)*. To date, several PGPRs have been recognized for their exceptional ability to degrade ethylene and various toxic chemicals such as solvents, organic compounds, and agricultural inputs. These findings are supported by studies such as those conducted by *Backer et al. (2018)*. Indeed, to neutralize metal toxicity, PGPR can bind to negative functional groups in the cell wall, allowing interactions with positive metal ions (*Syed & Chinthala, 2015*). These authors reported in their study that *Bacillus* strains showed a significant level of lead biosorption (*Syed & Chinthala, 2015*). *Wani & Khan (2014)* also showed that *Bradyrhizobium* can be used to promote the growth of plants, as well as for the detoxification of heavy metals in soils polluted by these metals. *Jiang et al. (2017)* mentioned that high concentrations, rhizospheric bacteria such as *Bacillus, Pseudomonas, Cupriavidus*, and *Acinetobacter* showed tolerance to metals such as Pb, Cd, and Cu.

## PGPR, application in maize and cowpea cultivation

### PGPR in the improvement of maize cultivation

Maize (*Zea mays* L.) is one of the world's most widely cultivated annual tropical herbaceous plants. It is a staple food for many populations because of its nutritional value in food and feed and its use in industry (*Revilla et al., 2022*; *Rouf Shah, Prasad & Kumar, 2016*). To meet the rapidly increasing population, the need for maize will require highly sustainable production and resilient farming systems (*Shiferaw et al., 2011*). Huge amounts of chemical inputs are often used to increase crop yields, but the overuse of these chemicals has negative environmental consequences. In this context, using PGPR for its beneficial effects on agricultural production and environmental preservation is a better alternative for sustainable agriculture. Indeed, several studies have proven the beneficial effects of PGPR on the germination, growth, and yield of cereal crops (*Agbodjato et al., 2016*; *Amogou et al., 2018*). According to studies by *Breedt, Labuschagne & Coutinho (2017)*, the use of *Paenibacillus alvei, Bacillus pumilus, Bacillus safensis*, and *Brevundimonas vesicularis* strains as bioinoculants resulted in maize yield improvements ranging from 24% to 34%. Due to their specific mechanisms of action, PGPRs have the ability to fix atmospheric nitrogen ($N_2$) and delay nitrogen remobilization in maize plants, resulting in increased crop yields (*Kuan et al., 2016*). These authors explain that this remobilization of Nitrogen in plants is mainly correlated with plant senescence leading to high maize cobs of

**Table 1 Mechanism(s) involved in PGPR strains and their effects on maize plants.**

| Species | Mechanism(s) involved | Results | References |
|---|---|---|---|
| *Serratia marcescens* | Promote better resistance of maize plants and nutrient acquisitions | Increased growth and yield parameters | *Amogou et al. (2019)* |
| *Sinorhizobium* sp. A15, *Bacillus* sp. A28, *Sphingomonas* sp. A55 and *Enterobacter* sp. P24 | Rhizobacteria populations are increasing both in terms of abundance and species diversity. | Improved growth leading to a 22–29% increase in maize yield, | *Chen et al. (2021)* |
| *Chryseobacterium* sp. NGB-29 and *Flavobacterium* sp. O NGB-31 | Nitrogen fixation and production of large amounts of IAA | Increased growth parameters | *Youseif (2018)* |
| *Bacillus subtilis* | Ammonia synthesis | Biofertilization | *Tahir et al. (2017)* |
| *Burkholderia cepacia* | Biocontrol and phosphate Solubilization | Increased length and surface area of leaves, shoots, roots and dry weight of plants | *Zhao et al. (2014)* |
| *Bacillus* sp. 12D6 *Enterobacter* sp 16i | Indole-3-acetic acid Salicylic acid | Significant improvement in root length, number of root tips and surface area | *Jochum et al. (2019)* |
| *Proteus penneri (Pp1)*, *P. aeruginosa (Pa2)*, *Alcaligenes faecalis (AF3)* | EPS production | Improve leaf area, and plant biomass | *Naseem et al. (2018)* |
| *B. subtilis* DHK | IAA production promotes transcription of the ACC synthase enzyme. Alleviating drought stress in maize | Significant increase in plant biomass. Increased antioxidant enzyme activity and reduced reactive oxygen species | *Sood, Kaushal & Sharma (2020)* |
| *Stenotrophomonas chelatiphaga* LPM-5 | Amylase synthesis | Significantly increased weight gain, iron (Fe) content of roots and shoots | *Ghavami et al. (2017)* |
| *Azotobacter chroococcum* | Biofertilization and Bioprotection | Soils become healthier and nutrient yields in maize improve. | *Song et al. (2021)* |

up to 30.9% with reduced nitrogen (N) fertilizer application. Several authors have reported the positive effect of PGPR on the growth of maize (*Adoko et al., 2020*; *Agbodjato et al., 2015*; *Amogou et al., 2019*). Research by *Pereira et al. (2011)* also explored the effect of bioprotection by PGPRs on maize crops. Indeed, maize is often confronted with the worrying presence of the toxigenic fungus Fusarium. The application of certain PGPR strains, such as *Microbacterium oleovorans* and *Bacillus amyloliquefaciens*, in the form of corn seed coatings, demonstrated effective protection against the pathogenic fungus *Fusarium verticillioides* (*Pereira et al., 2011*). In a similar study conducted by *Chen et al. (2021)* in China, it was shown that inoculation of different PGPR strains such as *Sinorhizobium* sp. A15, *Bacillus* sp. A28, *Enterobacter* sp. P24 and *Sphingomonas* sp. A55 significantly impacted on maize growth and resulted in an increase in grain yield of 22–29%. The research carried out by *Ghavami et al. (2017)* revealed the potential of bacterial strains in mitigating drought stress in maize. Indeed, their study showed that inoculation of the *Bacillus* DHK strain with maize plants significantly increased plant biomass. Compared to the control group, the use of this strain led to a reduction in reactive oxygen species and an increase in antioxidant enzyme activity (*Ghavami et al., 2017*). In their study, *Song et al. (2021)* showed that inoculation with *Azotobacter chroococcum* promotes plant nutrient uptake and improves maize crop yield without negatively impacting soil stability. In their work, *Marulanda et al. (2010)* observed that inoculation of

**Table 2 Mechanism(s) involved in PGPR strains and their effects on cowpea plants.**

| Species | Mechanism(s) involved | Results | Plant(s) | References |
|---|---|---|---|---|
| *Bacillus subtilis Dcl1* | Biocontrol and abiotic stress abatement agent | Improved cowpea growth and resistance to biotic and abiotic stress factors. | Cowpea (*Vigna unguiculata*) | *Jayakumar, Nair & Radhakrishnan (2021)* |
| *B. cereus* | Lipopeptides | Boosts bioprotection | Bean (*Phaseolus* vulgaris) | *Hashami et al. (2019)* |
| *Bacillus* spp. | Induction of disease resistance in cowpea against mosaic disease | Increase seed germination and seedling vigor; protection of plants against the pathogen | Cowpea (*Vigna unguiculata*) | *Shankar Udaya et al. (2009)* |
| *Pseudomonas libanensis TR1 Rhizophagus irregularis* | Biofertilization | Significant increases in shoot dry weight and in the number of pods and seeds per plant and grain yield | Cowpea (*Vigna unguiculata*) | *Rocha et al. (2019)* |
| *B. subtilis Fcl1* | Pesticide toxicity alleviating and growth-promoting impact | Improvement in cowpea growth and also toxicity alleviating effects of pesticide | Cowpea (*Vigna unguiculata* | *Juby et al. (2021)* |
| *Bradyrhizobium sp., B. megaterium, B. circulans, P. fluorescens* | Biofertilization and biological control | Fertilization and bio-control management; Increased growth and reduced the amount of chemical fertilization NPK | Cowpea (*Vigna unguiculata*) | *Zaghloul et al. (2019)* |

*Bacillus* megaterium into maize roots enhanced the ability of roots to absorb water in the presence of saline conditions. The Table 1 presented the Mechanism(s) involved in PGPR strains and their effects maize plants.

### PGPR in the improvement of cowpea cultivation

*Vigna unguiculata* L., commonly known as cowpea, is a legume widely appreciated for its nutritional value widespread consumption, and considerable economic importance worldwide. It is one of the most important legumes in the world, containing excellent nutritional factors while offering several agronomic, environmental, and economic benefits. The nutritional values of the cowpea have been widely reported. It is exceptionally rich in several nutrients, particularly high protein content, antioxidants, polyunsaturated fatty acids, polyphenols, and dietary fibers (*da Silva et al., 2018*; *Gonçalves et al., 2016*).

To promote healthy and sustainable agriculture, PGPR are attracting more attention from researchers and agricultural promoters due to their beneficial effects in improving agricultural production and protecting crops in soils affected by biotic and abiotic stress. In their study, *Udaya Shankar et al. (2009)* demonstrated the efficacy of *Bacillus* spp. strains in protecting cowpea plants against the common bean mosaic virus (CBMV). These strains also improved cowpea plant growth by inducing resistance against CBMV, under laboratory and field conditions. *Zaghloul et al. (2019)* showed that the PGPR strains used are good for fertilization management and bio-control which reduce the amount of NPK chemical fertilization. The research work conducted by *Rocha et al. (2019)* revealed that cowpea fields could perform better if PGPR seed and AM fungal isolates were used in low-input farming systems. The work of *Jayakumar, Nair & Radhakrishnan (2021)*

showed that identifying the properties of *B. subtilis* Dcl1 and demonstrating its probiotic effect on the plant *Vigna unguiculata* confirmed the potentialities of this strain as a biofertilizer, biocontrol, and bioremediation agent to improve agricultural productivity. Also, the study carried out by *Juby et al. (2021)* on cowpea has shown the beneficial application of the *Bacillus* Fcl1 endophyte to promote cowpea plant growth, even in the presence of toxic pesticides. In addition, these researchers demonstrated that the *Bacillus* Fcl1 strain can be used as a biological agent to mitigate the undesirable effects of toxic pesticides on plants. Table 2 presented the Mechanism(s) involved in PGPR strains and their effects on cowpea plants.

### Importance of PGPR in maize and cowpea cultivation

The use of PGPR-based biological formulations as a substitute for chemical fertilizers and pesticides in maize and cowpea cultivation offers significant advantages for several reasons. These include promoting environmental sustainability, reducing production costs, and improving crop yields. Indeed, PGPRs are beneficial micro-organisms, and in view of their aforementioned mechanisms of action, their use can considerably reduce dependence on chemical fertilizers, which have harmful effects on the environment, such as soil and water pollution, heavy metal contamination of crops, and so on. Thus, growing maize and cowpea with PGPRs will improve yields, resist plant pathogens, preserve ecosystems, and mitigate the effects of climate change on the environment, increasing crop resilience. By using PGPRs, farmers can reduce their dependence on expensive chemical fertilizers that are sometimes inaccessible to all strata in rural areas. In addition, the use of PGPRs promotes the utilization of nutrients present in the soil, thereby optimizing yields and reducing expenditure on agricultural inputs. By using PGPRs in maize and cowpea cultivation, we can obtain higher-quality crops, with higher nutrient content and no heavy metals. This can help increase the market value of crops and satisfy consumers' growing demands for quality products. This is a result of their ability to stimulate the plant immune system and boost resistance to environmental stresses such as drought, disease and pests, the use of PGPRs can enable farmers to reduce crop losses and ensure more sustainable agricultural production.

## THE CURRENT MARKET FOR PGPR PRODUCTS

In response to global population growth, it is imperative to sustainably increase global agricultural production while seeking solutions to reduce or eliminate excessive chemical use and combat abiotic stress and pathogens. According to the work of *Arora, Mehnaz & Balestrini (2016)*, biological formulations of products designed to promote plant growth, improve soil fertility, and combat plant pathogens offer ecological alternatives to traditional agrochemicals. In recent years, we see many natural products, including biofertilizers or biostimulants, to promote healthy and sustainable agriculture. Biofertilizers are generally considered to be products containing living micro-organisms that can be applied to seeds, plants or soil to promote plant growth. *Marcel et al. (2021)* also explained the difference between biofertilizers and biostimulants in their study. A

biofertilizer can be defined as a product containing bacterial or fungal inoculants that improve the availability and use of nutrients by plants. Biostimulants are compounds or microorganisms that enhance natural processes to improve nutrient uptake, stress tolerance, and crop quality (*Marcel et al., 2021*). *Du Jardin (2015)* reported that biofertilizers are included in biostimulants and biostimulants are broader than biofertilizers. These biofertilizers can also be called microbial biostimulants. Biofertilizers are commonly formulated from microorganisms beneficial to plant growth, and grouped into three main categories: arbuscular mycorrhizal fungi (AMF) (*Agbodjato et al., 2022*; *Jeffries et al., 2003*), plant growth-promoting rhizobacteria (PGPR) (*Agbodjato et al., 2021*; *Podile, Kishore & Gnanamanickam, 2006*) and nitrogen-fixing rhizobia (*Franche, Lindström & Elmerich, 2009*). Biofertilizers containing PGPR have the potential to enhance plant growth and yields by enriching the soil with nutrients and improving its quality (*de Andrade et al., 2023*). The first commercial formulations of PGPRs were patented and marketed by *Nobbe & Hiltner (1896)*, named Nitragin® biofertilizer made from Rhizobium. The global biofertilizers market is growing due to the increasing acceptance by scientists and farmers for its benefits in managing soil nutrients while ensuring sustainable agricultural production compared to chemical fertilizers which are not cost-effective. However, biofertilizers represent a relatively small share of the global market for synthetic agrochemicals (*Timmusk et al., 2017*). Among biofertilizers, those based on nitrogen fixers dominate the global market (*Soumare et al., 2020*). The biofertilizer market spans different regions of the globe, including Europe, North America, Asia-Pacific, Latin America, the Middle East, and Africa. In terms of revenues generated by biofertilizer production, North America (comprising mainly the USA, Canada, and Mexico) leads the global biofertilizer market, followed by Europe (Germany, UK, Spain, Italy, Hungary, and France) and Asia-Pacific (China, Japan, India, Australia, New Zealand, and other Asian countries). In 2017, revenues generated by biofertilizer markets totaled USD 495 million in North America, USD 450 million in Europe, USD 284 million in Asia-Pacific, USD 240 million in South America, and USD 44 million in Africa (*Soumare et al., 2020*).

Previous statistics have shown that Africa's market for biofertilizers is very small. Despite the advantages of biofertilizers, the African continent needs to take up the challenge of developing the biofertilizer market in Africa. To enhance the value of the African biofertilizer market, African governments and private sectors need to put in place sustainable extension and investment strategies to produce market and use quality biofertilizers by African farmers.

## CHALLENGES AND CONSTRAINTS: SCALABILITY AND COST-EFFECTIVENESS OF PGPRS FOR SUSTAINABLE ADOPTION IN AGRICULTURE

PGPR-based strategies are increasingly being studied and implemented in agriculture because of their potential to improve plant growth, reduce the use of chemical fertilizers, and promote sustainable farming practices. The scalability and profitability of PGPR-based strategies depend on several factors, including adaptability to different crops, farming

practices and environments, production costs, accessibility to farmers, ongoing research, and cost reduction.

A strategy based on PGPRs needs to be adapted to the specific needs of different crops, variable environmental conditions, and farming practices. These factors can make such strategies more complex, as they need to be adjusted to suit different crops, regions, and farming practices. It should also be added that strategies based on PGPRs must also take into account compatibility with farming practices, as farmers need to be able to integrate these strategies more easily into their cropping systems without significantly disrupting their routines. It is, therefore, necessary to involve them from the outset of the research to take account of their opinions in the implementation of these strategies.

PGPRs may require upfront costs for the isolation, production, and application of PGPR bacteria. Scalability will depend on the ability to make these costs affordable for farmers. In terms of accessibility, it should be noted that it is essential for farmers to have easy access to PGPRs and the knowledge needed to use them effectively. Efforts must be made to make these products and information accessible to farmers, particularly in developing regions.

To maximize the scalability and profitability of these strategies, collaboration between researchers, farmers, governments, and industry is essential to develop cost-effective, sustainable products and practices.

## CONCLUSIONS AND PROSPECT

The challenge of ensuring food and health security for the ever-growing population obliges us to turn to sustainable agriculture. PGPR is now considered to be an adequate substitute for chemical-based agricultural inputs. They are, therefore, very promising and bring sustainable benefits to agriculture. PGPR can maximally contribute to the provision of nutrients needed by the plant and to the protection of the plants. The diversity of activities offered by PGPRs in directly or indirectly promoting plant growth and yield improves soil physical quality, fertility, and functioning. The use of PGPR as biofertilizers or biostimulants in agriculture would therefore be one of the most promising solutions for sustainable and ecological agricultural production in the perspective of reducing the excessive use of chemical fertilizers and pesticides. Although there are some obstacles to the large-scale production and commercialization of biological formulations based on soil microorganisms, this biotechnology remains a promising alternative to promote in agricultural systems.

Given the importance of these PGPRs, it is vital to continue exploring these beneficial bacteria present in the rhizosphere, their reciprocal relationships, and their symbiosis with other plants to assess their potential for crop improvement and also to discover the means needed to maximize their effectiveness in agricultural production. Although the use of PGPRs offers considerable potential for improving sustainable agriculture, it is not without its challenges and limitations. Ongoing research, stakeholder education and adaptation to local conditions are essential to overcome these obstacles and maximize the benefits of PGPRs.

### Funding
This work was supported by the National Research Foundation, South Africa Grant (UID123634) to OOB, the North-West University Postdoctoral support to Nadège Adoukè Agbodjato. The funders had no role in study design, data collection and analysis, decision to publish, or preparation of the manuscript.

### Grant Disclosures
The following grant information was disclosed by the authors:
National Research Foundation, South Africa Grant: UID123634.
North-West University Postdoctoral Support to Nadège Adoukè Agbodjato.

### Competing Interests
The authors declare that they have no competing interests.

### Author Contributions
- Nadège Adoukè Agbodjato conceived and designed the experiments, performed the experiments, analyzed the data, prepared figures and/or tables, authored or reviewed drafts of the article, and approved the final draft.
- Olubukola Oluranti Babalola conceived and designed the experiments, performed the experiments, authored or reviewed drafts of the article, and approved the final draft.

### Data Availability
This is a literature review article.

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
