# Peer review of "Promoting sustainable agriculture by exploiting plant growth-promoting rhizobacteria (PGPR) to improve maize and cowpea crops"

_PeerJ, doi:10.7717/peerj.16836_

## Round 0.1 · original submission · Major Revisions

The authors are requested to revise a manuscript following reviewer comments. Authors should thoroughly read and understand each reviewer's comments, suggestions, and concerns. Clearly indicate the changes made in response to the reviewers' comments.

**Language Note:** The review process has identified that the English language must be improved. PeerJ can provide language editing services - please contact us at [email protected] for pricing (be sure to provide your manuscript number and title). Alternatively, you should make your own arrangements to improve the language quality and provide details in your response letter. – PeerJ Staff

Reviewer 1 ·

Basic reporting

The manuscript “Role of plant growth-promoting rhizobacteria (PGPR) in maize and cowpea cultivation: an effective alternative to promote sustainable agriculture” deals with a comprehensive overview of the potential benefits of using plant growth-promoting rhizobacteria (PGPR) as an alternative to conventional farming practices in the context of maize and cowpea production in Africa. Please find the further comments below.

Comment
• LN 20-23: Please rewrite it.
• The authors provide an excellent introduction to the topic, but it would be helpful to explicitly state the review's scope and limitations. What specific geographic regions or farming systems within Africa are the focus of this review? By providing this context, readers will be able to understand how the findings might be applied.
• In the abstract section write the future impact in one line.
• LN 42: I think this line is not that important, kindly remove it.
• LN 44: Kinldy check for typographical error, for example the right bracket in LN 44.
• LN 53: Please remove the old reference and update with the latest reference.
• LN 69: Please recheck the data of soil salinity.
• How do the costs of PGPR compare to traditional inputs like fertilizers and pesticides? Including this information would help assess the practicality and economic viability of PGPR adoption.
• According to the paper, previous studies have demonstrated the benefits of PGPRs, but it fails to provide specific examples or references to support these claims. Specific case studies or experiments that demonstrate the effectiveness of PGPRs in maize and cowpea production would strengthen the argument.
• As the paper suggests PGPR as an alternative to conventional farming practices, it would be valuable to discuss the scalability and cost-effectiveness of PGPR-based strategies. Are there any barriers to large-scale production and distribution of PGPR?
• LN 82: Please revisit the aims and objectives of the manuscript.
• LN 143-150: Please rewrite the line.
• Section “Rhizodeposition and its interaction with PGPR in the rhizosphere” needs to be elaborated and discussed.
• A conclusion that summarizes the key findings and provides practical recommendations for farmers, policymakers, and researchers interested in adopting PGPR-based strategies would be helpful to the paper. Identifying future areas for research would also help guide further research into optimizing PGPR application and addressing any remaining knowledge gaps.

Experimental design

Please see the comments above

Validity of the findings

Please see the comments above

Additional comments

Please see the comments above

Reviewer 2 ·

Basic reporting

1. Please improve the title and make it short and crisp
2. The role of PGPR should be highlighted in the abstract itself, more specifically the major mechanisms and associated beneficial impacts.
3. Completely rephrase the abstract and focus on the major conclusions and future thrust drawn from this literary review.
4. Some of the sentences have no relevance to the main objective of this study for eg; “In addition, Reductions in yields of wheat, maize, rice, barley, etc. are direct consequences of 65 climate change due to the frequency and severity of high temperatures and drought (Carmen and 66 Roberto, 2011). In addition, soil salinity is one of the world's major environmental challenges, 67 transforming fertile land into unproductive land at a rate estimated at around 1-2% per year in 68 arid regions. Soil salinization has rendered around 7% of the planet's land infertile, representing 69 around 20% of the total arable land surface (Rasool et al., 2013).”
5. Please restructure the whole content and write in a more scientific manner. Write now it looks like a book chapter.
6. What is the correlation of Figure 1 with this study? Please improve this figure in a more conceptual manner.
7. Table 1 and 2: many recent and crucial examples are missing and those must be included. Additionally, the mechanism part is superficially written.
8. Heavy metal stress is also a kind of abiotic stress so it’s better to improve the legends.
9. Focus on the examples of PGPRs and success stories to mitigate various stresses.

Experimental design

Ideally the following content should be incorporated in a structured manner.
1. Introduction 1.1 Background and Overview 1.2 Objectives of the Review
2. Plant Growth-Promoting Rhizobacteria (PGPR): Definition and Characteristics 2.1 PGPR Mechanisms in Promoting Plant Growth 2.2 Importance of PGPR in Sustainable Agriculture
3. PGPR Application in Maize Cultivation 3.1 Effects of PGPR on Maize Growth and Development 3.2 Influence of PGPR on Nutrient Uptake in Maize 3.3 Enhanced Resistance to Diseases and Pests in Maize 3.4 PGPR as a Biofertilizer for Maize Cultivation
4. PGPR Application in Cowpea Cultivation 4.1 Effects of PGPR on Cowpea Growth and Yield 4.2 Improving Nutrient Acquisition in Cowpea through PGPR 4.3 Mitigating Disease and Pest Pressure in Cowpea with PGPR 4.4 PGPR as a Biofertilizer for Cowpea Cultivation
5. Comparative Analysis of PGPR Benefits in Maize and Cowpea Cultivation 5.1 Similarities in PGPR Effects on Maize and Cowpea 5.2 Contrasting Responses in Maize and Cowpea to PGPR Treatment
6. Environmental and Economic Implications of PGPR Application 6.1 Reduced Environmental Impact with PGPR 6.2 Economic Benefits of PGPR in Sustainable Agriculture
7. Challenges and Limitations 7.1 Factors Affecting PGPR Efficacy 7.2 Potential Risks and Constraints in PGPR Implementation
8. Future Prospects and Recommendations 8.1 Expanding PGPR Research in Agriculture 8.2 Strategies for Successful PGPR Integration 8.3 Promoting Adoption of PGPR in Farming Practices

Validity of the findings

Please improve based on aforementioned comments.

·

Basic reporting

The introduction of review should focus on targeted subject and provide updated information.
There are various repetitions of information in various headings; the author should exclude such repeated information
Headings on the impact of pesticides should be removed; add soil pollution issues in introduction to highlight the need and significance of PGPR
Sections related to rhizosphere and rhizodeposition should merge into a single section for instance, Rhizosphere microbiome
A major part of the review focused on PGPR characteristics and general action mechanisms. it should include the beneficial impact of PGPR as biofertilizer
An economic analysis of the utilization of PGPR as biofertilizer
Also, the various approaches to employing PGPR for soil and plant health
The above-mentioned point should be included in review to make it interesting and as a meaningful contribution in this field

Experimental design

The author selected valuable literature to review; however, published research in 2023 is not well cited , it would be better to add works from the current year to make your data update with time.

Validity of the findings

There is valid support and knowledge about PGPR capabilities and work modes, but factors biological, economical and environmental that make these PGPR as biofertilizer can be added.

Additional comments

Conclusively, the review requires covering the missing areas about approach and benefits of PGPR as biofertilization.er
How would their application method work in the field?
What would be the economic or biological benefits to the farmer and market ?
How would this approach augment the process to achieve the SDGs about food and malnutrition?

---

## Round 0.2 · accepted · Accept

The manuscript is improved and now it is acceptable in its current form.

Reviewer 1 ·

Basic reporting

The author made significant changes in the manuscript.

Experimental design

See section 1

Validity of the findings

See section 1

Additional comments

See section 1